:ᅌᐧ PLOS | ONE

# An evaluation of genetic causes and environmental risks for bilateral optic atrophy

**Andrew T. Chen** [ID][1], **Lauren Brady**[1], **Dennis E. Bulman**[2☯], **Arun N. E. Sundaram**[3☯], **Amadeo R. Rodriguez**[4☯], **Edward Margolin**[5☯], **John S. Waye**[6☯], **Mark A. Tarnopolsky**[1]*

**1** Department of Pediatrics, McMaster University, Hamilton, ON, Canada, **2** Children's Hospital of Eastern Ontario Research Institute, University of Ottawa, Ottawa, ON, Canada, **3** Department of Ophthalmology and Vision Sciences, Sunnybrook Health Sciences Center, University of Toronto, Toronto, ON, Canada, **4** Department of Ophthalmology, McMaster University, Hamilton, ON, Canada, **5** University of Toronto Department of Ophthalmology and Visual Sciences, Mount Sinai Hospital, Toronto, ON, Canada, **6** Department of Pathology and Molecular Medicine, McMaster University, Hamilton, ON, Canada

☯ These authors contributed equally to this work.
* tarnopol@mcmaster.ca

## Abstract

### Purpose

To assess the clinical utility of next-generation sequencing (NGS) for the diagnosis of patients with optic atrophy (OA).

### Design

Retrospective cohort study.

### Methods

97 patients were referred to the McMaster University Medical Center (Hamilton, Ontario) for evaluation of bilateral OA. All patients were sent for NGS including a 22 nuclear gene panel and/or complete mitochondrial DNA (mtDNA) sequencing. Positive genetic test results and abnormal vibration sensation were compared in patients +/- environmental exposures or a family history.

### Results

19/94 (20.2%) had a positive nuclear variant, of which 15/19 (78.9%) were in the *OPA1* gene. No positive mtDNA variants were identified. The detection of a positive genetic variant was significantly different in patients who reported excessive ethanol use, but not in patients who smoke (0/19 (0%) vs. 19/78 (24.4%), P = 0.0164 and 4/22 (18.2%) vs. 15/74 (20.3%), P = 0.829, respectively). Patients with a positive family history were more likely to have a positive genetic variant compared to patients with a negative family history (P = 0.0112). There were significantly more excessive drinkers with an abnormal vibration sensation (P = 0.026), and with a similar trend in smokers (P = 0.074).

**Data Availability Statement:** All relevant data are within the manuscript and its Supporting Information files.

**Funding:** Dr. Tarnopolsky's (MT) work on mitochondrial disorders is supported by kind donations from Mr. Warren Lammert and family, Mr. Dan Wright and family and MitoCanada (https://mitocanada.org/). Dr. Tarnopolsky's research into genetic disorders is supported by a grant from the Canadian Institute for Health Research Grant Number 143325 (http://www.cihr-irsc.gc.ca/e/193.html). The funders had no role in study design, data collection and analysis, decision to publish, or preparation of the manuscript.

**Competing interests:** The authors have declared that no competing interests exist.

## Conclusions

All positive genetic variants were identified in nuclear genes. We identified a potential independent pathophysiological link between a history of excessive ethanol consumption and bilateral OA. Further investigations should evaluate and identify potential environmental risk factors for OA.

## Introduction

Inherited optic neuropathies are genetic disorders characterized by the slow loss of the retinal ganglion cells (RGCs) and their axons that form the optic nerve [1]. Each optic nerve carries visual information from retinal photoreceptors to the brain. Clinically, patients with inherited optic neuropathies have a distinct presentation: they almost invariably have very symmetric visual loss without the presence of relative afferent pupillary defect which is present in almost all other causes of optic neuropathy and often have characteristic visual field defects (central or cecocentral scotomas). As with any optic neuropathy, optic atrophy (OA) is the end result of the insult affecting an optic nerve and tends to be symmetric in inherited optic neuropathies. It is defined as variable loss of the optic nerve axons and subsequent optic nerve pallor visible on examination with variable deterioration of central visual acuity [2].

The two most common forms of inherited optic neuropathy are autosomal dominant optic atrophy (DOA) secondary to pathogenic variants within *OPA1* (OMIM: 165500), and maternally inherited Leber hereditary optic neuropathy (LHON) (OMIM: 535000) [3]. OPA1 is a ubiquitous GTPase that localizes to the mitochondria and functions in mitochondrial fusion, cristae integrity, and mitochondrial DNA (mtDNA) maintenance [4,5]. Three primary mtDNA variants, m.11778G>A, m.3460G>A, and m.14484T>C, account for ~95% of all cases of LHON [6]. All three variants affect complex I subunits of the respiratory chain, resulting in impaired oxidative phosphorylation and increased levels of reactive oxygen species (ROS) [7,8]. Rarer cases of OA can be caused by pathogenic variants in the nuclear-encoded mitochondrial genes such as *WFS1*, *MFN2*, *POLG*, and *ACO2* [9].

DOA typically has an insidious onset during the first two decades of life and presents clinically with an incomplete penetrance and heterogenous clinical expressivity [10]. In contrast, LHON presents with subacute visual loss in one eye followed usually within 1–2 months by visual loss in the opposite eye. Visual loss usually occurs in the second and third decades of life and men are disproportionally affected for reasons that are not entirely clear [11,12]. Powerful molecular diagnostic tools, such as next-generation sequencing (NGS), are now being used in the work-up of individuals who present with optic atrophy.

In addition to hereditary causes, OA may also occur after exposure to environmental factors including: smoking, trauma, industrial toxins, and psychological stress [13]. Some of these exposures can trigger vision loss in LHON mutations [13,14], leading to our hypothesis that other environmental factors such as excessive ethanol use could unmask a potential underlying gene mutation associated with OA. It is well documented that methanol poisoning causes OA [15–17]; however, the excessive consumption of ethanol in association with OA remains unclear either as an independent toxin [18], or a second hit [13,19]. DNA repair pathways may have a role in protecting against DNA damage upon excessive ethanol consumption [20]. Vulnerability of the papillomacular bundle to mitochondrial dysfunction is a likely final etiological common pathway in most cases of OA due to epigenetic disorders [21], and this is further reinforced by the observation that all cases of LHON involve mtDNA genes and most cases of

OA in association with nuclear mutations occur within genes encoding for mitochondrial proteins.

In this study, NGS was used in the workup of individuals referred to a Canadian neurometabolic clinic for evaluation of OA. Testing included a panel of 22 nuclear genes associated with inherited optic neuropathies as well as sequencing of the entire mitochondrial genome. The main purpose of the current study was to determine the proportion of patients with pathogenic mutations in mtDNA or select nuclear genes associated with optic atrophy. A second aim was to determine whether excessive ethanol use and/or smoking were enriched in the mutation positive (as a trigger or second hit), or negative (independent etiology) patients.

## Methods

### Patients

This retrospective chart review involved patients who were assessed at the Neuromuscular and Neurometabolic Clinic in the McMaster University Medical Centre located in Hamilton, Ontario between 2013 and 2019. Approval for the study was obtained by written consent through the Hamilton Integrated Research Ethics Board (Research Ethics Board #2018-5519-C). The Hamilton Integrated Research Ethics Board approved the retrieval of data with knowledge of the identifying information.

All patients were referred for genetic testing after they were first identified by a neuroophthalmologist to have findings of optic nerve pallor or atrophy, with or without visual loss. All but two patients (# 84 and 87 with unilateral OA) had bilateral OA. All patients had a MRI scan of the brain and orbits, and were screened with the testing of vitamin B12, folate, anti-nuclear antibodies, anti-neutrophil cytoplasmic antibodies, angiotensin converting enzyme, syphilis screening, and anti-extractable nuclear antigen, which are the other potential causes of symmetric optic neuropathies. Only patients without a suspected deficiency (i.e. folate, B12) and with no evidence of multiple sclerosis, neuromyelitis optica, or other identified cause for OA were sent for genetic testing. Sixteen patients underwent nuclear DNA sequencing only, three patients underwent mtDNA sequencing only, and 78 patients underwent both. In total, 97 patients who underwent mtDNA and/or nuclear DNA sequencing were reviewed (Fig 1). Patients were referred for mtDNA and/or nuclear DNA genetic sequencing based upon likelihood of variant identification by the referring specialist. Patients were of differing ethnicities and the median age at last evaluation was 38 years. Five patients were children (5.2%) and 92 patients were adults (94.8%). The average age of onset was 28 years (range, 3–79).

Each individual was asked about a family history of vision loss or optic atrophy during their consultation appointment. All patients were classified as having a positive, negative, or uncertain family history.

### Mitochondrial DNA sequencing

mtDNA sequencing and deletion analysis was performed primarily on leukocyte-derived (and sometimes muscle-derived) mitochondrial DNA by Next Generation Sequencing (NGS) at McMaster University Medical Centre Molecular Genetics Laboratory. The entire mitochondrial genome was sequenced using the Nimblegen's SeqCap EZ Mitochondrial Genome Design. Typically, the mean depth of coverage was >10,000x. Deletions and duplications of sequences within the mitochondrial genome were assessed by normalizing the depth of coverage at each nucleotide against data obtained from samples known to have the normal mtDNA structure [22]. Patients with a detected variant at a heteroplasmy level less than 5% were considered to have a negative test result. All remaining results ≥ 5% heteroplasmy were classified

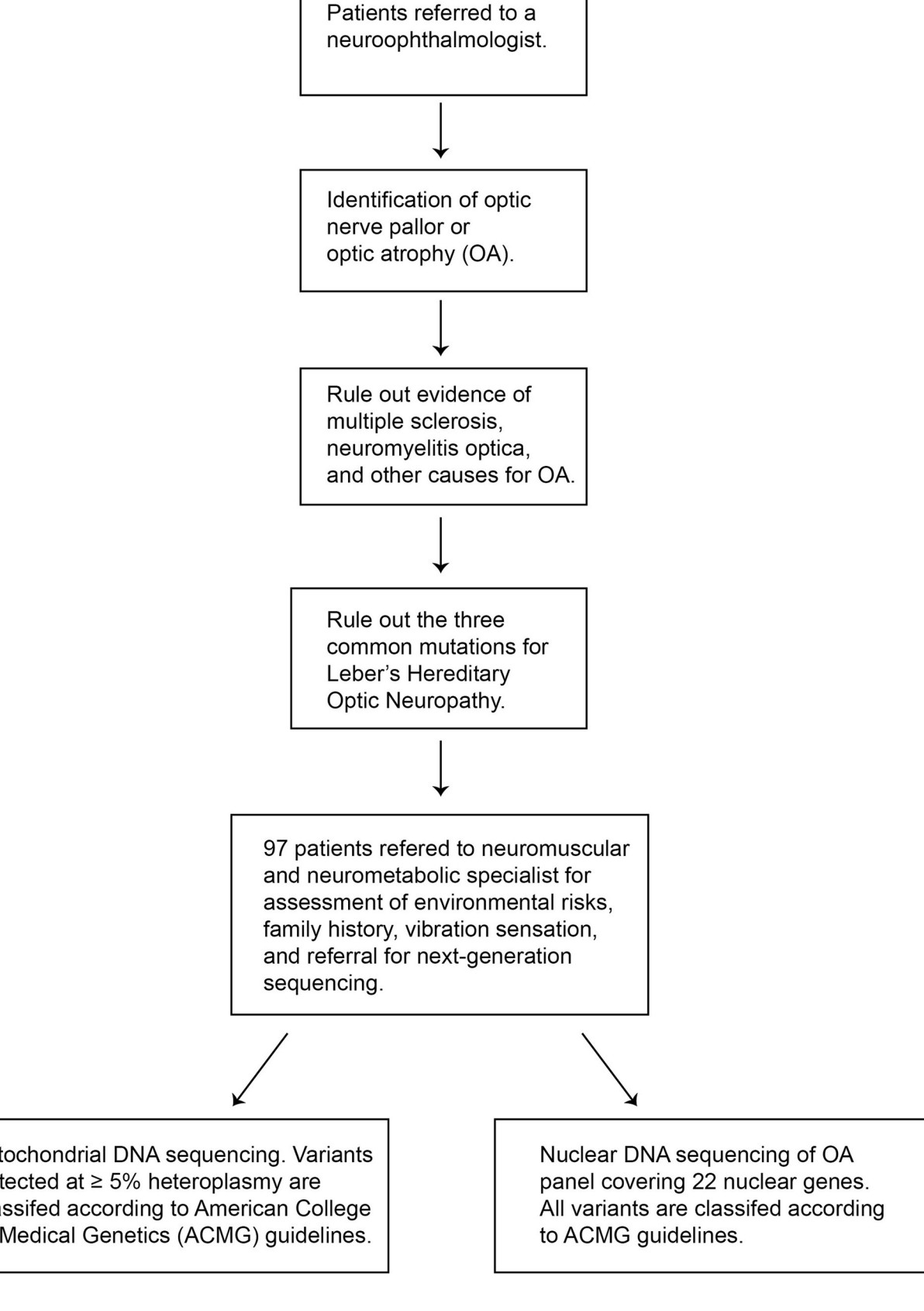

**Fig 1. Flowchart of the retrospective study.** Patients were referred for genetic testing after extensive workup by referring specialists.

according to the American College of Medical Genetics (ACMG) guidelines for variant interpretation [23].

## Nuclear DNA sequencing

Nuclear gene testing was performed on blood derived leukocytes at the Children's Hospital of Eastern Ontario (CHEO)/Newborn Screening Ontario (NSO). The NGS "Progressive External Ophthalmoplegia (PEO)/Optic Atrophy" panel covered 22 nuclear genes: *ACO2*, *AUH*, *TWNK (C10orf2)*, *C12orf65*, *CISD2*, *MFN2*, *MGME1*, *MTPAP*, *NDUFS1*, *OPA1*, *OPA3*, *POLG*, *POLG2*, *RRM2B*, *SLC19A3*, *SLC25A4*, *SPG7*, *TIMM8A*, *TK2*, *TMEM126A*, *TYMP*, and *WFS1*. The panel was created and consistently revised in partnership between Newborn Screening Ontario (NSO), London Health Science Center, and Hamilton Health Sciences to test for genes currently known to be associated with OA. CHEO/NSO followed standard procedures for DNA sample preparation and quality control. Library construction was accomplished with Kapa Hyper Prep library kits. Standard manufacturer protocols were followed to perform targeted capture with Nimblegen's SeqCap EZ Choice. The coding exons, as well as splice junctions (50 base pair (bp) padding), of the genes in the panel were targeted. Sequencing was done with 150 bp paired-end reads on an Illumina MiSeq, which generates an average of 800 Mbp of sequence for each sample. Mean coverage design and sequencing workflow can consistently cover >99.5% of target coding bases at >30X read depth (>99.7% at >20X), with 20X coverage as their minimal quality standard for reporting. All patients with a detected nuclear variant were reported and classified according to the ACMG guidelines for variant interpretations.

## Variant classification

GATK Best Practices were followed throughout the workflow. For preprocessing, prior to alignment, sequence reads were automatically demultiplexed, adaptors were trimmed, and those sequences that did not meet the default settings on MiSeq Reporter (v2.5) were removed. The reads were then aligned to hg19 human reference genome with BWA-mem options -a -M (v0.7.12-r1039). Duplicate reads were marked by Picard 1.134 and excluded from downstream analysis. GATK 3.5 modules for indel realignment and base recalibration were calibrated to known single nucleotide polymorphisms (SNPs)/indels (eg. dbSNP138 and 1000G gold standard). Variant calling was performed per sample by GATK 3.5 HaplotypeCaller at each CCDS exon with a 10 bp padding. Hard filters for SNPs and indels were applied with default settings to the variant call set. Annovar was used to annotate the variant call set. Tertiary analyses were performed at PierianDX (St. Louis, MO). Variants were filtered initially with minor allele frequency < 3%. Prioritization was given to variants with known ClinVar pathogenicity or those predicted to alter the protein's function.

Variant classification was following 2015 ACMG Standard and guidelines for the interpretation of sequence variants [23]. A result was considered "Positive" if the variant was classified as "Pathogenic" or "Likely Pathogenic". Family studies to assist in the interpretation of the genetic variants identified were performed depending on the availability of informative family members. Heterozygous "Variants of Uncertain Significance" (VUS) were reported for genes associated with an autosomal dominant phenotype. Heterozygous VUS in genes associated with autosomal recessive inheritance were not reported in the absence of a second possible variant.

## Environmental exposure assessment

Heavy ethanol use was defined as more than one drink per day or 10 drinks per week for women and two drinks per day or 15 drinks per week for men [24]. Binge drinking was defined as four drinks for women and five drinks for men in about two hours [24]. Individuals were classified as excessive drinkers if they reported heavy drinking or binge drinking within two years prior to, or at the time of, vision loss or diagnosis of optic atrophy. Smoking status of patients were assessed and classified as "Smoker" if they reported smoking in the past or currently and "Never smoker" if they did not report any history of smoking.

## Neurological examination including vibration sensation testing

Given that there is precedent for the *OPA1* [25,26] and some LHON variants [27] to be associated with extra-ophthalmological neurological manifestations [28], a complete neurological examination was carried out in all patients by a single neuromuscular/neurometabolic specialist (MT) to determine whether there was evidence of neurological impairment in addition to the documented OA. The only neurological finding (other than optic atrophy) present in > 10% of the cohort was a reduction in vibration sensation (128Hz tuning fork), and that was graded and recorded using an ordinal scale (0 = normal; 1 = mild; 2 = moderate; 3 = severe reduction; and 4 = completely absent), with any result other than 0 classified as abnormal vibration sensation.

## Statistical tests

To test if a positive environmental exposure risk factor was enriched in mutation positive or negative patients, we compared the proportion of patients that had a positive genetic test with a positive or negative history of environmental exposure (drinking/smoking) using a two-sided chi-squared test. Given there is *a priori* prediction for there to be greater number of individuals with a positive family history to have a positive genetic variant, a one-sided chi-squared test was used to compare the proportion of patients that had a positive genetic test with a positive or negative family history. The latter test was used in comparing abnormal neurological examination findings in patients with a positive or negative history of environmental exposure (drinking/smoking).

Significance was determined at $p < 0.05$, 95% CI. All statistics are done using Graphpad Prism.

# Results

## Nuclear variants

Ninety-four patients in the cohort had genetic testing through the Optic Atrophy Nuclear Gene panel (Fig 2A, S1 Table). Pathogenic or likely pathogenic variants were identified in 22/94 (23.4%). One patient and his mother (patients # 1 and 67) were identified to have a heterozygous likely pathogenic variant in the *ACO2* gene, which previously has only been reported with an autosomal recessive pattern of inheritance [29]. No second variant in this gene was identified. One patient (# 85) was identified to have a heterozygous likely pathogenic variant in the *AUH* [30,31]. No second variant in this gene was identified. Therefore, these latter three patients were considered to have negative testing. Consequently, a positive genetic test felt to likely explain the OA phenotype was identified in 19/94 (20.2%). As expected, the highest proportion of positive tests were found in the *OPA1* gene (15/19 (78.9%)). The remaining four positive tests were identified in *ACO2* (patient # 30) and *WFS1* (patient # 8, 77, and 97) (Fig 2C).

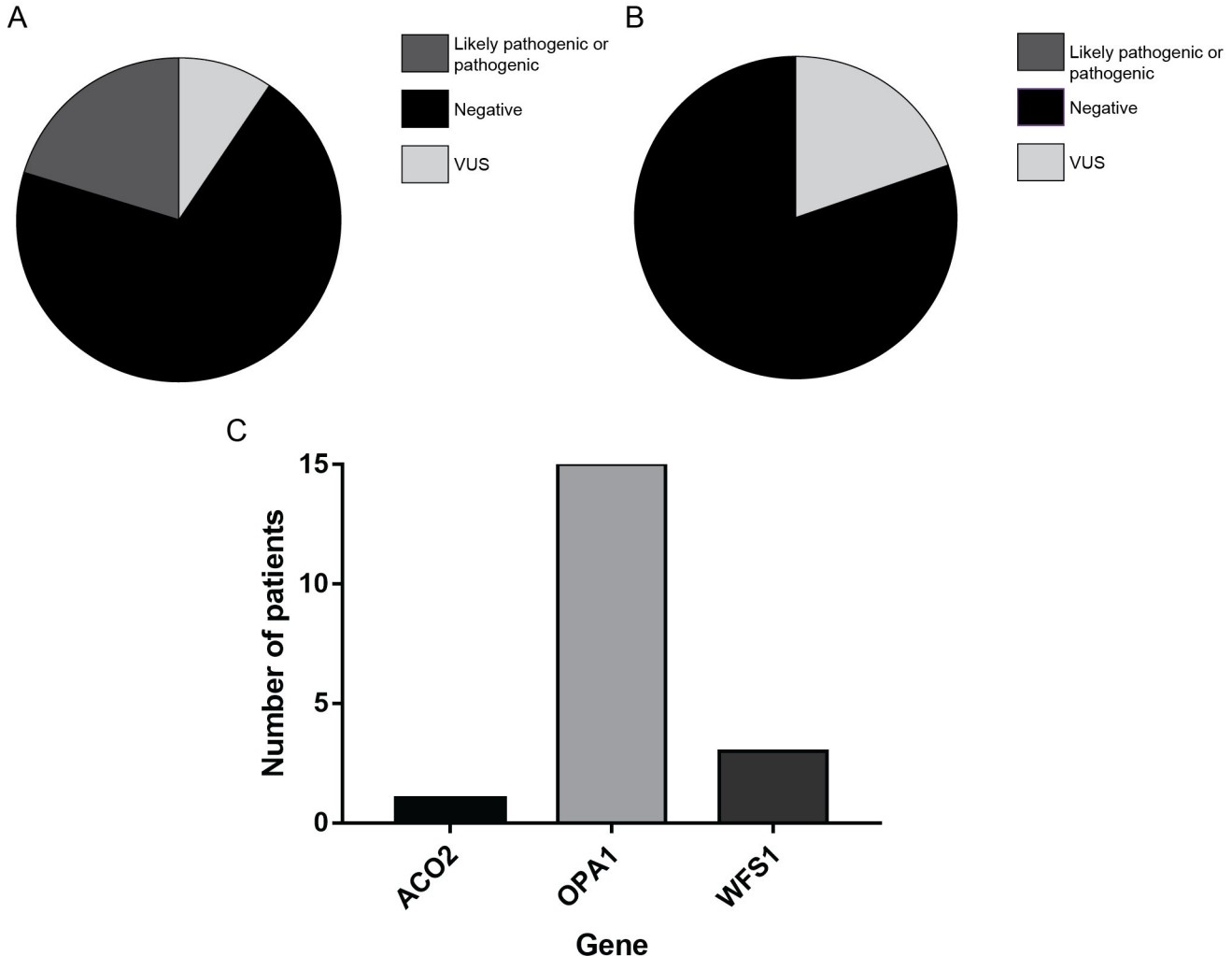

**Fig 2. Variant identification and classification of ninety-seven patients sent for NGS.** (A) PEO/Optic Atrophy nuclear panel results (N = 94). (B) mtDNA results (N = 81). (C) Frequency of patients with a definitive positive molecular diagnosis with causative nuclear gene.

Variants of uncertain significance (VUS) possibly associated with OA were identified in 9/94 (9.6%) of the patients in this cohort. Overall, 68/94 (70.2%) of the patients in this cohort had a negative genetic test result (Fig 2A).

## Mitochondrial variants

Eighty-one patients underwent mtDNA sequencing (Fig 2B, S1 Table). We identified no pathogenic or likely pathogenic mtDNA variants in leukocyte-derived DNA associated with OA and as a result, no patients had a positive genetic test result; however, VUS were identified in 16/81(19.8%). Therefore, 65/81 (80.2%) had a negative mtDNA result (Fig 2B).

## Family history

In the patients with a positive family history of vision loss or optic atrophy, 10/30 (33.3%) had a positive genetic test result. In contrast, there were significantly fewer (P = 0.011) positive genetic test results in patients with a negative or uncertain family history (9/67 (13.4%)) (Fig 3).

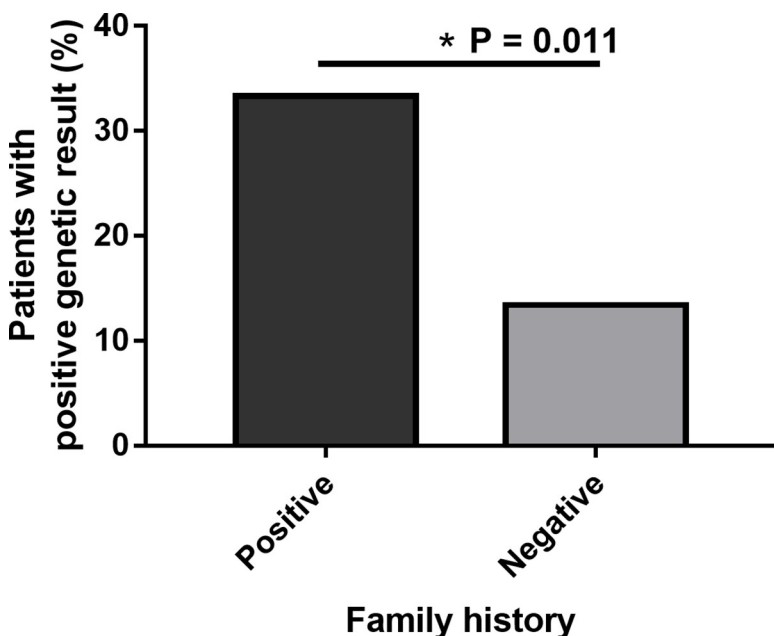

**Fig 3. Positive predictive value of a genetic test result in patients with a positive or negative family history of optic atrophy.** Of those that reported a positive family history, 10/30 (33.3%) patients had a positive genetic test result. Of those with a negative family history, 9/67 (13.4%) had a positive genetic test result (OR, 1.20–9.02; *, p = 0.011; CI = 95%).

### Excessive ethanol use

Nineteen patients (19/97) were identified in the cohort to have a history of excessive ethanol consumption. Of the 19 who reported a history of excessive ethanol consumption, none had a positive genetic test result (0/19). In contrast, of those individuals who did not report a history of excessive ethanol consumption, there was a significantly greater (P = 0.016) number of patients (19/78 (24.4%)) who had a positive genetic test result (Fig 4A).

### Smoking

Of the 22 patients who reported smoking in the past or currently, 4/22 (18.2%) patients had a positive genetic test result. Of those that had never smoked, there is not a significantly different number of patients that have a positive genetic test result (15/74 (20.3%), (P = 0.829)) (Fig 4B).

### Neurological examination and vibration sensation

Other than optic disc pallor/OA, the only abnormal neurological examination finding present in > 10% of the cohort was an abnormal vibration sensation (20.6%). Of these patients, the proportion of patients with a positive history of excessive ethanol use or smoking that demonstrated an abnormal vibration sensation test was compared to the proportion of patients with a negative history that demonstrated an abnormal vibration sensation test. There were significantly more patients (P = 0.026) with an abnormal vibration sensation test who drank excessively (7/19 (36.8%)) as compared to those who did not drink excessively (13/78 (16.7%) (Fig 5A). Although not statistically significant, there was a trend (P = 0.074) for a greater percentage of patients who smoked to have an abnormal vibration sensation test (7/22 (31.8%)) compared to those who did not smoke (13/74 (17.6%)) (Fig 5B).

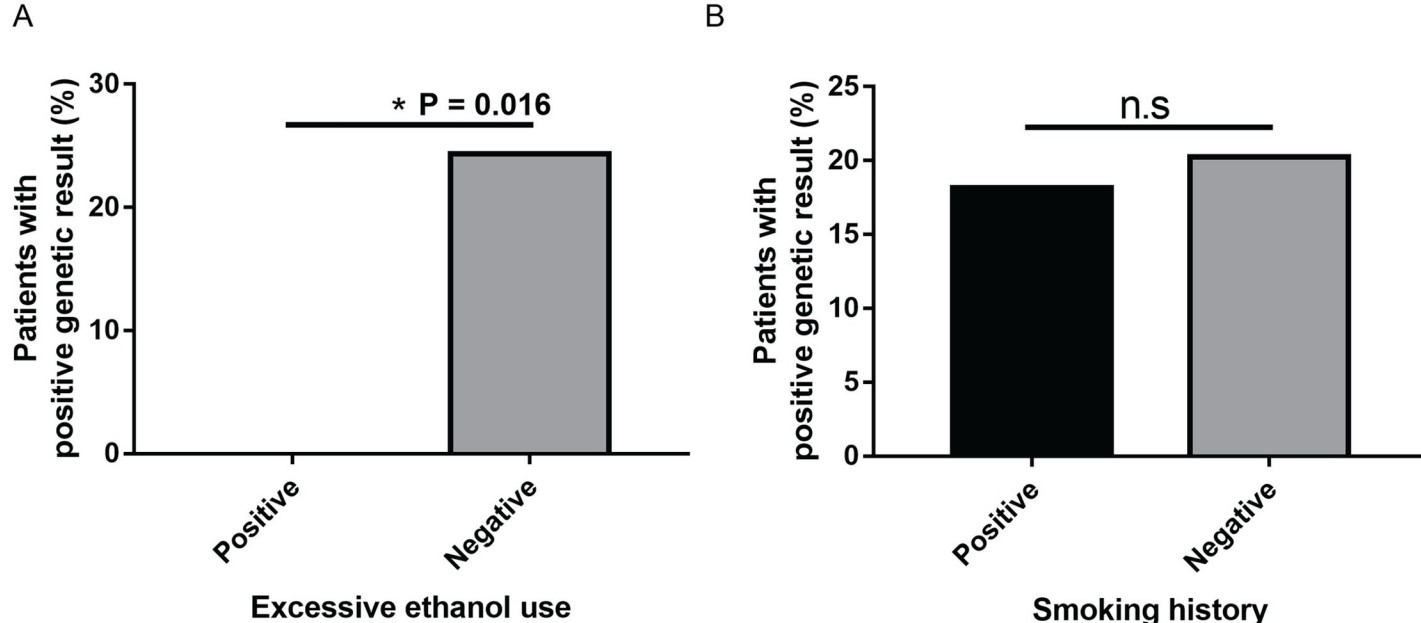

**Fig 4. Relationship between excessive ethanol use or smoking and the detection of a likely pathogenic or pathogenic variant.** (A) Percentage of patients who have a likely pathogenic or pathogenic variant given that they are excessive drinkers or are not excessive drinkers. Patients that report excessive ethanol consumption have a lower chance of possessing a positive genetic variant, whereas those that do not excessively drink have a higher positive detection rate (24.4%) (OR, 0–0.70; *, p = 0.016; CI = 95%). (B) Percentage of patients who reported to have smoked in the past/currently or have never smoked that have a likely pathogenic or pathogenic variant (OR, 0.29–2.78; p = 0.829; CI = 95%).

## Discussion

The first aim of this review was to investigate the diagnostic utility and spectrum of mutations identified using an NGS-based OA nuclear panel and complete mtDNA sequencing in a cohort of patients found to have OA with or without vision loss. In this report, all positive genetic tests detected by NGS were in nuclear genes. Furthermore, 15/19 (78.9%) of the positive genetic test results detected in the nuclear genes were in *OPA1*. As reported previously, these results support that most cases of inherited optic atrophy can be attributed largely to dominant optic atrophy resulting from genetic changes in *OPA1* (50–80%) [32–34]. Most of the individuals reviewed experienced an irreversible loss of visual acuity, where central vision loss tended to manifest first. The neurological examination was normal in all *OPA1* patients in this cohort and did not identify any of the extra-ophthalmological manifestations reported in some *OPA1* mutation positive patients [35].

Pathogenic or likely pathogenic variants in the *WFS1* and *ACO2* genes were found in a smaller proportion of our OA patients. Mutations in *WFS1* are usually associated with Wolfram syndrome (OMIM# 222300), sometimes known as DIDMOAD (diabetes insipidus, diabetes mellitus, optic atrophy, and deafness) [36]. Wolfram syndrome is characterized by juvenile diabetes mellitus, hearing loss, and optic nerve atrophy [37]. No patients in our study with *WFS1* variants presented with *bona fide* Wolfram syndrome. In our cohort of patients, biallelic mutations in *WFS1* resulted in isolated OA (with or without vision loss) in patients # 8 and 97. Patient # 8 presented with asymptomatic optic atrophy (correctable vision loss) and was identified to have a pathogenic variant (*WFS1* c.1941C>A; p.Cys647) [38,39] and a VUS (c.1597C>T; p.Pro533Ser) *in trans* (Table 1) [40]. Patient # 97 had bilateral vision loss and was homozygous for a missense variant (c.1672C>T; p.Arg558Cys) [41,42]. At 33 and 39 years of age, respectively, neither individual had diabetes nor hearing loss, two of the canonical features

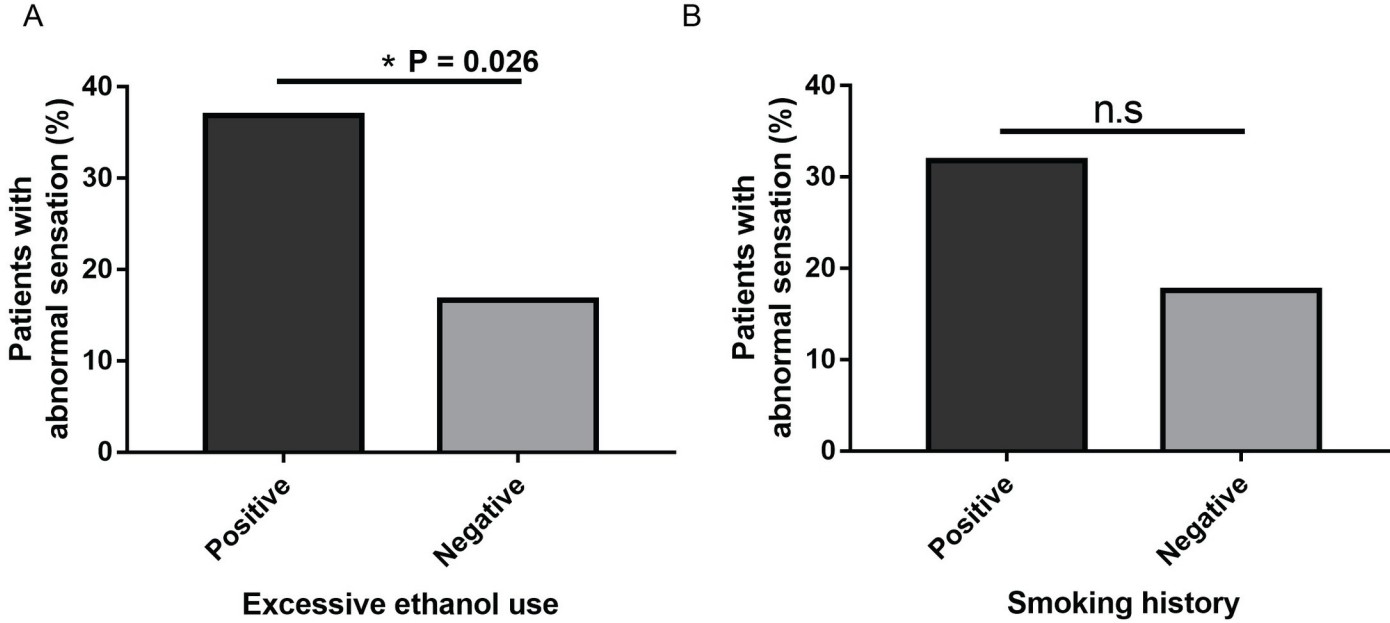

**Fig 5. Comparison of patients with drinking or smoking history in relation to performance on vibration sensation test.** (A) Percentage of patients that drink or do not drink who have abnormal vibration sensation (OR, 1.02–8.68; *, p = 0.026; CI = 95%). (B) Percentage of patients that smoke or do not smoke who have an abnormal vibration sensation (OR, 0.80–6.06; p = 0.074; CI = 95%).

of Wolfram syndrome. These two cases support recent findings that biallelic heterozygous or homozygous *WFS1* variants can be associated with nonsyndromic or syndromic autosomal recessive OA without diabetes or hearing loss [41]. In contrast, two other patients (patients # 16 and 77) had biallelic *WFS1* variants and presented with OA but also had type 2 diabetes mellitus (T2DM) and a positive family history of hearing loss, respectively. Family members of both patients were unavailable to phase the variants. While patient # 16 was asymptomatic (correctable vision loss), patient # 77 did experience a loss of visual acuity because of their OA. The *WFS1* variants detected in patient # 16 were c.1682T>G (p.Ile561Ser) [40] and c.1936_1943del (p.Phe646Valfs*63) [43], which has been associated with nonsyndromic OA with T2DM. Patient # 77 had a pathogenic *WFS1* variant (c.1309G>C; p.Gly437Arg) and a likely pathogenic variant (c.2020G>A; p.Gly674Arg) that have both been previously documented in cases of autosomal recessive Wolfram [41,44]. Surprisingly, this patient did not present with the other canonical feature of Wolfram syndrome (T2DM). Altogether, these results add to the wide spectrum of mutations and resulting phenotypes in patients with *WFS1* variants.

**Table 1. Positive genetic testing of recessive variants likely to explain the OA phenotype.**

| Patient No. | Gene | DNA | | Protein | |
|---|---|---|---|---|---|
| | | Allele 1 | Allele 2 | Allele 1 | Allele 2 |
| 8 | WFS1 | c.1941C>A | c.1597C>T | p.Cys647* | p.Pro533Ser |
| 16 | WFS1 | c.1682T>G | c.1936_1943del | p.Ile561Ser | p.Phe646Valfs*63 |
| 77 | WFS1 | c.1309G>C | c.2020G>A | p.Gly437Arg | p.Gly674Arg |
| 97 | WFS1 | c.1672C>T | c.1672C>T | p.Arg558Cys | p.Arg558Cys |
| 30 | ACO2 | c.220C>G | c.2086G>A | p.Leu74Val | p.Glu696Lys |

Optic Atrophy Type 9 (OPA9) is associated with biallelic pathogenic variants in *ACO2* [29]. The ACO2 protein is a tricarboxylic acid (TCA) enzyme that interconverts citrate to isocitrate. Only nine families with *ACO2*-related visual loss have been documented in the literature [29,45–49]. Our finding of a likely pathogenic variant (c.220C>G; p.Leu74Val) and VUS (c.2086G>A; p.Glu696Lys) *in trans* in patient # 30, suggests that this represents another case of an association between OA and *ACO2* mutations. The *ACO2* variant (c.220C>G; p.Leu74-Val) has been reported to be associated with cellular respiration defects [49], and reduced protein expression [29], in OA. As previously reported by others [29], our patients had a relatively early age at onset (5y) of OA symptoms, suggesting that OA associated TCA cycle defects present earlier than the more common *OPA1* mutations.

In LHON patients, the three primary mutations account for approximately 95% of all cases [50]. The most common mutation at nucleotide position m.11778 accounts for 60–90% of all cases [51]. However, many sporadic cases of idiopathic bilateral symmetric OA have been documented and found to harbour pathogenic mtDNA point mutations [52]. No patients in this study were found to have any of the three common LHON variants, nor any other likely pathogenic or pathogenic variants. This is likely due to the fact that patients who presented with LHON phenotype were already tested for the three most common mutations by referring physicians and thus not sent for further genetic testing in our clinic. Variants of uncertain significance were identified in the mtDNA of 16 individuals, but none were felt to be pathogenic based upon many criteria (heteroplasmy, high prevalence in public data bases, benign prediction model program score, etc.). Consequently, it is very unlikely that any of the VUS found in mtDNA had an etiological link to OA. Given that neither of the three primary LHON mutations [50], the rarer LHON variants, nor any other potentially pathogenic mtDNA variants were found in our cohort of 81 patients, it questions the utility of complete mtDNA sequencing in cases of OA. Our data suggests that mtDNA sequencing may only be needed if a patient reports a family history of vision loss suspicious for mitochondrial inheritance pattern or presents with canonical LHON onset: semi-acute fairly profound visual loss in one eye accompanied by mild optic nerve head swelling which is followed by visual loss in the opposite eye within days to months.

DOA and LHON are both inherited optic neuropathies and genetic testing should be considered in the context of personal and family history. Expectedly, patients who reported a family history of OA were more likely to have a positive genetic test result than those with a negative family history (33.3% vs. 13.4%, P = 0.011) (Fig 3). This demonstrates the importance of obtaining an accurate family history prior to ordering genetic testing [53], but also suggests that the absence of a family history should not preclude patients from getting genetic testing.

Nevertheless, the genetic results from this study were positive for 20.2% patients through the nuclear gene panel and for no patients through mtDNA sequencing. The relatively modest number of positive results from the nuclear gene panel may suggest the presence of genetic changes in rare or candidate genes not on the panel, variants in intronic regions, epigenetic factors, or the presence of other variants not readily identified using NGS (i.e., trinucleotide repeats, large deletions, etc.). It is also possible that there is no Mendelian genetic etiology behind these cases of optic atrophy.

The second aim of this review was to investigate the role of environmental factors, such as excessive ethanol consumption and/or smoking, in the development of OA. Our cohort was evaluated to see if these environmental factors could precipitate OA as a second hit in gene positive patients or be an independent etiology for OA in gene negative patients. In contrast to our *a priori* hypothesis, none of the patients who reported excessive ethanol use (N = 19) had a pathogenic or likely pathogenic variant. In contrast, of the remaining individuals who did not report excessive ethanol use, 19/78 (24.4%) had a positive genetic test result. Given that

excessive ethanol use was enriched in our gene negative patients, (P = 0.016) our study indirectly supports an independent pathophysiological link between excessive ethanol consumption and the development of OA. The association between ethanol consumption and OA was also supported by anecdotal evidence in our cohort. For example, one patient (patient #42) woke up after one night of binge drinking with a complete bilateral lost of vision. In contrast with the strong pathophysiological link between methanol consumption and the development of OA, the evidence that excessive ethanol consumption *per se* can lead to OA is less clear [15–17]. For example, a cross-sectional study of 100 optic atrophy patients reported only one chronic alcoholic patient with primary OA [54]. Other case-control [55] and *in vivo* [56] studies have implicated a possible link between excessive ethanol consumption and OA. Our study suggests it is likely that environmental toxins, such as chronic ethanol consumption, can be an independent cause for the development of OA. However, due to our relatively small sample size of excessive drinkers (N = 19), it remains important even for patients who are presymptomatic carriers of a mitochondrial variant or Mendelian condition associated with DOA to not drink [13,57–59].

Contrary to excessive ethanol use, the difference in the proportion of patients with positive genetic results between smokers and never smokers was not significant (Fig 4). Thus, we could neither conclude that smoking *per se* was an independent risk factor for OA, nor that it was an exacerbating factor. In contrast, there is strong evidence that smoking exacerbates LHON symptoms through an increase in the production of ROS and further uncoupling of the respiratory chain [60]. Given that we did not find any positive LHON mtDNA variants in our cohort, we could not evaluate a causal relationship with smoking and LHON.

Lastly, we hypothesized that peripheral neurological defects would be more prevalent in the patients who consumed excessive amounts of ethanol compared to those that do not. Our results suggest that excessive ethanol consumption, even in the absence of a positive genetic variant(s), could be associated with greater extra-ophthalmological neurological manifestations. In our cohort of 19 patients who excessively drank, a significantly greater proportion (P = 0.026) had abnormal vibration sensation compared to patients who did not excessively drink (36.8 vs. 16.7%). This further suggests that ethanol use can independently cause OA given that excessive ethanol consumption was associated with a propensity for neurotoxicity, either large fiber sensory nerve and/or dorsal column impairment. One potential confounding variable is the fact that the excessive ethanol consumers were significantly older than the those who do not drink excessively (age at last evaluation 59.6y vs. 40.6y). In contrast, the influence of smoking on extra-ophthalmological neurological manifestations in our cohort is inconclusive but with a trend suggesting that vibration abnormalities may be associated with a positive smoking history (P = 0.074). Further studies will be needed to further understand the potential neurological effects of smoking in patients with optic neuropathies.

In summary, optic atrophy is a debilitating condition that can result from a variety of genetic and non-genetic factors. Mutations in *OPA1* are the most commonly identified genetic cause of OA in our cohort. This case series also supports previous observations that variants in *WFS1* can be associated with non-syndromic OA instead of Wolfram syndrome. Complete mtDNA sequencing appears to be of limited utility in patients with isolated OA and a clinical history not suggestive of LHON. The uniformly negative result with the mtDNA sequencing likely reflects the selection cohort where patients were referred for OA and not a specific history typical of LHON. Our data also suggests that excessive ethanol consumption, but not smoking, may be an isolated cause or risk factor for OA. There is a need for further research in understanding the genetic and environmental factors contributing to the development and progression of optic atrophy and vision loss.

## Supporting information

**S1 Table. The summary of 97 patients assessed at the McMaster University Medical Center for optic atrophy.** All patients were sent for nuclear and/or mitochondrial sequencing. Variants were classified according to the ACMG guidelines for variant classification. Smoking, excessive ethanol use, type of vision loss, and extra features are reported and described. (XLSX)

## Author Contributions

**Conceptualization:** Lauren Brady, Mark A. Tarnopolsky.

**Data curation:** Andrew T. Chen, Lauren Brady, Dennis E. Bulman, Arun N. E. Sundaram, Amadeo R. Rodriguez, Edward Margolin, John S. Waye, Mark A. Tarnopolsky.

**Formal analysis:** Andrew T. Chen, Lauren Brady, Dennis E. Bulman, Arun N. E. Sundaram, Amadeo R. Rodriguez, Edward Margolin, John S. Waye, Mark A. Tarnopolsky.

**Funding acquisition:** Mark A. Tarnopolsky.

**Investigation:** Andrew T. Chen, Dennis E. Bulman, Arun N. E. Sundaram, Amadeo R. Rodriguez, Edward Margolin, John S. Waye, Mark A. Tarnopolsky.

**Methodology:** Andrew T. Chen, Lauren Brady, Dennis E. Bulman, John S. Waye, Mark A. Tarnopolsky.

**Project administration:** Mark A. Tarnopolsky.

**Supervision:** Lauren Brady, Mark A. Tarnopolsky.

**Writing – original draft:** Andrew T. Chen, Lauren Brady, Mark A. Tarnopolsky.

**Writing – review & editing:** Andrew T. Chen, Lauren Brady, Dennis E. Bulman, Arun N. E. Sundaram, Amadeo R. Rodriguez, Edward Margolin, John S. Waye, Mark A. Tarnopolsky.

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
