## [Decision Letter · Decision Letter 0]

17 Oct 2019

PONE-D-19-26430

An evaluation of genetic causes and environmental risks for bilateral optic atrophy

PLOS ONE

Dear Dr. Tarnopolsky,

Thank you for submitting your manuscript to PLOS ONE. After careful consideration, we feel that it has merit but does not fully meet PLOS ONE’s publication criteria as it currently stands. Therefore, we invite you to submit a revised version of the manuscript that addresses the points raised during the review process.

A flow chart with of the process from NGS to variant selection would be easily understandable. In addition to that, please read carefully the comments from the reviewer and address them point by point and with any additional experiments if necessary.

We would appreciate receiving your revised manuscript by Dec 01 2019 11:59PM. To enhance the reproducibility of your results, we recommend that if applicable you deposit your laboratory protocols in protocols.io, where a protocol can be assigned its own identifier (DOI) such that it can be cited independently in the future. For instructions see: http://journals.plos.org/plosone/s/submission-guidelines#loc-laboratory-protocols

We look forward to receiving your revised manuscript.

Kind regards,

Obul Reddy Bandapalli, MSc, PhD

Academic Editor

PLOS ONE

Journal Requirements:

2.  In the ethics statement in the manuscript and in the online submission form, please provide additional information about the patient records used in your retrospective study.

Specifically, please ensure that you have discussed whether all data were fully anonymized before you accessed them and/or whether the IRB or ethics committee waived the requirement for informed consent.

If patients provided informed written consent to have data from their medical records used in research, please include this information.

3. In your data file, please ensure you have removed any potentially identifying information, such as date of birth and specific age of onsent.

Reviewers' comments:

Reviewer's Responses to Questions

**Comments to the Author**

1. Is the manuscript technically sound, and do the data support the conclusions?

Reviewer #1: Yes

Reviewer #2: Yes

2. Has the statistical analysis been performed appropriately and rigorously? 

Reviewer #1: I Don't Know

Reviewer #2: Yes

3. Have the authors made all data underlying the findings in their manuscript fully available?

Reviewer #1: Yes

Reviewer #2: Yes

4. Is the manuscript presented in an intelligible fashion and written in standard English?

Reviewer #1: Yes

Reviewer #2: Yes

5. Review Comments to the Author

Reviewer #1: In this research article by Chen et al., 97 individuals with optic atrophy (OA) were analyzed by NGS of 22 nuclear genes and/or complete mitochondrial DNA (mtDNA) sequencing in order to assess the clinical utility of NGS for the diagnosis of OA; moreover the impact of environmental factors was investigated, identifying ethanol consumption as a potential independent risk study. The manuscript is overall well written and the results are fully explained. However, it is not clear to me what correlation the authors expected to exist between environmental factors such as ethanol consumption and genetics defects associated with OA, as the presence of a causative genetic mutation (and not a susceptibility factor) should be responsible for the development of the disease independently from environmental factors. Moreover, I am not sure that the definition of ethanol consumption as an epigenetic factor is appropriate, as the term epigenetic modification refers to the specific DNA modification (for example, DNA methylation)that can develop as a result of environmental agents; and still it’s not clear why the authors expected an epigenetic factor to interact with a known causative mutation. I think the authors should revise this section and make their hypothesis more understandable. Finally, it should also be clarified how exactly the genes in the panel were selected and why some patients underwent only nuclear genes or mitochondrial DNA sequencing instead of both.

Reviewer #2: Gist/Summary: The authors come up with analysis of variants in patients with optic atrophy and liaise this with environment factors. In this process, they sequence the mitochondrial genomes of the patients by employing a retrospective cohort design. Detecting positive variant in a cohort of samples in the patients with excessive alcohol ( ethanol) intake and smokers augurs well with the proposal. It is certainly the need of the hour for such diseases linking NGS and epigenetics!

Strengths: Well aimed with two diffident objectives for the design. The authors also deliberate well in evaluating the genetic causes and associating LHON and other neuropathies.

Weaknesses: A pictorial methodology would have been nice. The figures could have been bettered. A detailed discussion on associated gene risk , finding variants through epistatic effects could have been very nice!

Lines 69-77: The role of DNA repair genes and a sentence on it could be a good addition.

Lines 84-90: Pretty long sentence., could be split into two!

Lines 117-119: A Table with restrospective design, odds ration, criteria etcs. could be a value addition.

Lines 136-138: The panel details must be elaborated. There are a few open reading frames that are from known unknown regions/unannotated ( c10orf2, c12orf65) encompassing Twinkle helicase/PEO genes. Did the authors find any variants associated with them?

Something similar of this sort, whence presenting Table 1, SLC25A1 gene links WFS1 and Aconitase (ACO2) gene. Did the authors find any variants associated with them. A soft gene interaction map would be a good one ( https://www.ncbi.nlm.nih.gov/pmc/articles/PMC4746751/ )

Line 204: "Previously" mistyped.

The possibility of variants, be it homozygous dominant or heterozygous mutations in Metabolic syndromes cannot be ruled out. Could the authors discuss these aspects?

A subtle challenge in linking the statistical significance of their environmental study is the lack of inherent samples for ethanol/smoking. Could machine learning heuristics be an answer for this in the future as good number of datasets are available in sequence read archive.

6. PLOS authors have the option to publish the peer review history of their article (what does this mean?). If published, this will include your full peer review and any attached files.

Reviewer #1: No

Reviewer #2: Yes: Prashanth N Suravajhala

---

## [Author Response · Author response to Decision Letter 0]

31 Oct 2019

Obul Reddy Bandapalli, MSc, PhD

Academic Editor

PLOS ONE

Dear Dr. Bandapalli:

Thank you for considering our manuscript and allowing us to address the issues brought forth.

The manuscript has been edited to meet PLOS ONE’s style requirements. In particular, the “References” heading has been changed to the appropriate heading style 1. A caption for “Supporting Information” has been added after the references section. Table 1 has been correctly formatted to have no empty rows with a clearer presentation of information.

The ethics statement has been updated in the manuscript and in the online submission form: “The Hamilton Integrated Research Ethics Board approved the retrieval of data with knowledge of the identifying information.” In addition, any potential link to the patients have been removed – date of birth and specific age of onset have been removed from the S1 Table.

“Fig 1: Flowchart of the retrospective study” has been added to the manuscript to allow readers to understand the process of patient and variant selection for NGS. All in-text citations have been changed accordingly.

Further, we would like to address the comments from the editors in the order in which they were presented.

Sincerely,

Mark

Mark Tarnopolsky, MD, PhD,

Department of Pediatrics, McMaster University, 

Hamilton, Ontario, L8N 3Z5, CANADA.

Phone 905-521-2100 (76593)

FAX 905-577-8380

tarnopol@mcmaster.ca

Reviewer # 1:

We would like to thank this reviewer for his/her helpful comments and have made revisions along the lines suggested.

Specifically: We have changed “epigenetic factors” in the abstract and in our hypothesis to “environmental [risk] factors.” In addition to the changes in our hypothesis, we suggest in our discussion that environmental toxins such as chronic ethanol consumption can be an independent cause for the development of OA. With these changes in mind, we believe it is more clear that chronic ethanol consumption is an environmental factor that could be triggering epigenetic changes in known or unknown associated genes, rather than a definitive epigenetic factor for OA. Furthermore, statements have been added to clarify how patients were selected for mtDNA and/or nuclear NGS (lines 117-119). Lastly, a statement has been added to clarify the selection of genes in the PEO/Optic Atrophy nuclear panel (lines 144-146).

Reviewer # 2:

We would like to thank this reviewer for his/her helpful comments and have made revisions along the lines suggested.

Specifically: A pictorial methodology (Fig 1) has been added to describe the retrospective design, criteria, and variant classifications. A sentence on the potential role of DNA repair genes has been added (lines 86-87). The lines 84-90 in the unrevised manuscript has been split into two concise sentences. We have added a sentence to state how the panel was created and that all detected nuclear variants were reported. Therefore, no variants were discovered in SLC25A1 or the unknown regions/unannotated. The typo found on line 204 in the unrevised manuscript has been fixed. The possibility of a metabolic syndrome cannot be ruled out through our study, as stated in lines 88-91. All patients are tested for B12/folate deficiencies and other causes for OA, prior to NGS. The pictorial methodology added can help to clarify this. Lastly, machine learning heuristics may be an answer in the future for overcoming challenges in sampling, but it may be out of the scope of the current study.

---

## [Decision Letter · Decision Letter 1]

11 Nov 2019

An evaluation of genetic causes and environmental risks for bilateral optic atrophy

PONE-D-19-26430R1

Dear Dr. Tarnopolsky,

We are pleased to inform you that your manuscript has been judged scientifically suitable for publication and will be formally accepted for publication once it complies with all outstanding technical requirements.

With kind regards,

Obul Reddy Bandapalli, MSc, PhD

Academic Editor

PLOS ONE

Additional Editor Comments (optional):

Reviewers' comments:

Reviewer's Responses to Questions

**Comments to the Author**

1. If the authors have adequately addressed your comments raised in a previous round of review and you feel that this manuscript is now acceptable for publication, you may indicate that here to bypass the “Comments to the Author” section, enter your conflict of interest statement in the “Confidential to Editor” section, and submit your "Accept" recommendation.

Reviewer #1: (No Response)

Reviewer #2: All comments have been addressed

2. Is the manuscript technically sound, and do the data support the conclusions?

Reviewer #1: Yes

Reviewer #2: Yes

3. Has the statistical analysis been performed appropriately and rigorously? 

Reviewer #1: I Don't Know

Reviewer #2: Yes

4. Have the authors made all data underlying the findings in their manuscript fully available?

Reviewer #1: Yes

Reviewer #2: Yes

5. Is the manuscript presented in an intelligible fashion and written in standard English?

Reviewer #1: (No Response)

Reviewer #2: Yes

6. Review Comments to the Author

Reviewer #1: (No Response)

Reviewer #2: I am satisfied with all the changes rendered by the authors. They could have answered the responses in line, but it's okay.

7. PLOS authors have the option to publish the peer review history of their article (what does this mean?). If published, this will include your full peer review and any attached files.

Reviewer #1: No

Reviewer #2: Yes: Prashanth Suravajhala

---

## [Editor Report · Acceptance letter]

18 Nov 2019

PONE-D-19-26430R1 

An evaluation of genetic causes and environmental risks for bilateral optic atrophy 

Dear Dr. Tarnopolsky:

I am pleased to inform you that your manuscript has been deemed suitable for publication in PLOS ONE. Congratulations! Your manuscript is now with our production department. 

With kind regards,

on behalf of

Dr. Obul Reddy Bandapalli 

Academic Editor

PLOS ONE